# LITE: Modeling Environmental Ecosystems with Multimodal Large Language Models

**Haoran Li[‡], Junqi Liu[¶], Zexian Wang[§], Shiyuan Luo[†], Xiaowei Jia[†], Huaxiu Yao[*]**
[*]UNC-Chapel Hill, [‡]PolyU, [§]UMich, [¶]Independent Researcher, [†]University of Pittsburgh
haoran.li.cs@gmail.com, huaxiu@cs.unc.edu

## Abstract

The modeling of environmental ecosystems plays a pivotal role in the sustainable management of our planet. Accurate prediction of key environmental variables over space and time can aid in informed policy and decision-making, thus improving people's livelihood. Recently, deep learning-based methods have shown promise in modeling the spatial-temporal relationships for predicting environmental variables. However, these approaches often fall short in handling incomplete features and distribution shifts, which are commonly observed in environmental data due to the substantial cost of data collection and malfunctions in measuring instruments. To address these issues, we propose LITE – a multimodal large language model for environmental ecosystems modeling. Specifically, LITE unifies different environmental variables by transforming them into natural language descriptions and line graph images. Then, LITE utilizes unified encoders to capture spatial-temporal dynamics and correlations in different modalities. During this step, the incomplete features are imputed by a sparse Mixture-of-Experts framework, and the distribution shift is handled by incorporating multi-granularity information from past observations. Finally, guided by domain instructions, a language model is employed to fuse the multimodal representations for the prediction. Our experiments demonstrate that LITE significantly enhances performance in environmental spatial-temporal prediction across different domains compared to the best baseline, with a 41.25% reduction in prediction error. This justifies its effectiveness. Our data and code are available at https://github.com/hrlics/LITE.

## 1 Introduction

Sustainable management of environmental ecosystems has been gaining massive attention due to its impact on global climate, and food and water security (Tamburini et al., 2020; O'Donnell et al., 2023; Li et al., 2023). However, this task has become more challenging due to the rising demand for environmental services from the growing population, more frequent extreme weather events, and the climate change (Zurek et al., 2022; Jasechko et al., 2024). The modeling of environmental ecosystems is critical for the sustainable management, as it provides important information about the spatial-temporal dynamics of key physical variables. For example, accurate prediction of streamflow and water temperature for large river basins can aid in the decision-making for water resource allocation and water quality control. An agricultural monitoring system can help develop policies that maintain farming efficiency and sustainability, and stabilize economies in agriculture-intensive areas.

Physics-based models (Markstrom et al., 2015; Theurer et al., 1984; Zhou et al., 2021) and data-driven models (Zaremba et al., 2014; Hochreiter & Schmidhuber, 1997; Moshe et al., 2020; Jia et al., 2021) have been developed for spatial-temporal prediction in environmental applications. In particular, machine learning models (e.g., recurrent neural networks (Zaremba et al., 2014; Hochreiter & Schmidhuber, 1997) and graph neural network based models (Chen et al., 2021; Sun et al., 2021)) significantly improve the performance of environmental spatial-temporal prediction by modeling the complex spatial-temporal correlations automatically from observed data. Despite the improved predictive accuracy, two key challenges remain

unsolved. Firstly, existing methods exhibit vulnerability to incomplete features, which can be a common issue in environmental applications due to sensor failures and the high cost of field surveys needed to measure certain variables across different regions and time. Second, these prior approaches can be severely affected by data distribution shifts, which are commonly observed in environmental applications due to variations in human management and changes in weather conditions.

To address these challenges, we propose a general large language model (LLM)-based framework, termed LITE, to handle incomplete features and distribution shifts behind the environmental data. Specifically, we first transform the spatial-temporal (ST) data into semantic time-series and temporal trend images. The semantic time-series reflects inter-variable correlations while the temporal trend image depicts temporal dynamics of all variables in the form of line graph images. In this step, we tackle the incomplete features by replacing the absent variables with special token (e.g., [MASK]) in semantic time-series and linearly interpolating the temporal trend image. Then, LITE leverages a semantic time-series encoder and a vision encoder to jointly capture spatial-temporal dynamics and correlations. During this process, we incorporate multi-granularity information, i.e., merging previous observational data from different granularity such as week, month, and year to facilitate the handling of distribution shifts. Finally, the information, representing the spatial-temporal dynamics and dependencies, will be processed by a frozen LLM under the guidance of domain instructions, including dataset description, task description, and target statistics.

The primary contribution of this paper is the development of the new framework LITE, which introduces an innovative multimodal representation learning framework for modeling environmental ecosystems. This approach not only exploits learned knowledge within foundation models to handle the heterogeneity of different environmental domains, but also presents consistent robustness to different levels of incomplete observations and distribution shifts. We evaluate LITE on three environmental spatial-temporal prediction scenarios: streamflow, water temperature, and agricultural nitrous oxide ($N_2O$) emissions. For all scenarios, our empirical results demonstrate an improvement of 12.2%, 56.0%, and 55.6% in terms of RMSE, verifying the the effectiveness of LITE. Furthermore, we conduct experiments to demonstrate that the superiority of LITE remains or is even more significant under (1) data distribution shift; (2) extreme incomplete features, offering further justifications on the robustness and effectiveness of LITE.

## 2 Preliminaries

In this section, we introduce the notations used in this paper and formally define the problem setup.

**Definition 1 (Spatial-Temporal Series)**: For a region $r \in \mathcal{R}$ in an environmental ecosystem, we denote the current timestamp as $t$, and the time range as a set $\mathcal{T} = \{t - |\mathcal{T}| + 1, ..., t\}$ consisting of $|\mathcal{T}|$ evenly split non-overlapping time intervals. Then, the series of target physical variables $y_{r,t}$ (e.g., streamflow, water temperature, agricultural nitrous oxide ($N_2O$)) over multiple spatial locations is represented as

$$\mathcal{Y}_t = \{y_{r,t'} | t' \in \mathcal{T}, r \in \mathcal{R}\}. \tag{1}$$

**Problem Definition.** Given the input physical drivers $\mathcal{X} = \{x_{r,t'} | t' \in \mathcal{T}, r \in \mathcal{R}\}$ for multiple regions until the current time $t$, our goal is to predict the spatial-temporal series of the target variable $y_{r,t}$ until the current time, as

$$\mathcal{Y}_t = f_\theta(\mathcal{X}_t) \tag{2}$$

where $f_\theta$ represents the ST model parameterized by $\theta$, which predicts the target spatial-temporal series using input physical drivers. The input drivers are physical variables that can affect the state of target environmental ecosystems, e.g., meteorological and soil-related variables.

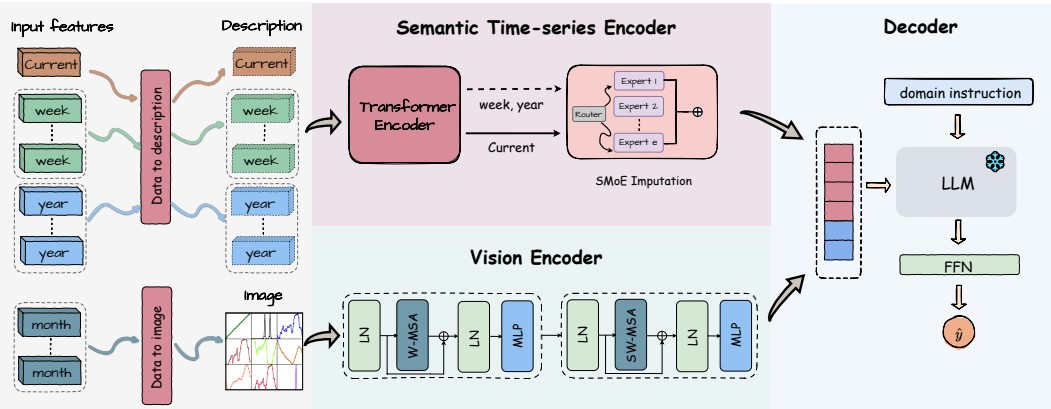

Figure 1: Overview of our proposed LITE model for environmental ecosystem modeling, which consists of (1) Transforming environmental data into natural language descriptions and line graph images; (2) multimodal representation learning; (3) multimodal fusion by LLM decoder.

## 3  Multimodal Large Language Model for Environmental Ecosystem Modeling

This section presents our proposed Multimodal **L**arge Language Model for Env**I**ronmen**T**al **E**cosystem Modeling (**LITE**). The goal of LITE is to capture ST dynamics and correlations from a multimodal perspective, and process this information within an LLM-based framework, as depicted in Figure 1. To elaborate, we first transform the ST data into semantic time-series data and temporal trend images. Incomplete features are addressed by replacing missing variables with a special token (e.g., [MASK]) in the semantic time-series, and by linearly interpolating in the temporal trend images. Subsequently, we utilize a semantic time-series encoder and a vision encoder to learn multimodal representations for the environmental ST data. To enhance the robustness to distribution shifts, we incorporate multi-granularity information from previous observations in this step. Finally, the multimodal representations are fused by a frozen LLM decoder to perform ST prediction under the guidance of domain instruction, i.e., dataset description, task description, and target statistics. Below, we detail each component of LITE.

### 3.1  Representation Learning from Semantic Time-Series

In this section, we first discuss how to learn representation from environmental data via semantic time-series, which includes three steps: (1) transforming environmental data into semantic time-series, (2) imputation of incomplete observations, and (3) multi-granularity information integration. We detail these steps below.

**Transforming Environmental Data into Semantic Time-Series.** Given the varying types and quantities of features across different environmental datasets, we propose to first unify these features and transform them into corresponding semantic descriptions. Specifically, we first format each data point $x_{r,t}$ of $K$ dimensions into a sequence of key-value pairs. For each pair, the key represents the feature description $c_{r,t}^k$, while the value is the corresponding feature value $x_{r,t}^k$. This can be formulated as follows:

$$\text{Linearize}(x_{r,t}) = \{[c_{r,t}^k : x_{r,t}^k]\}_{k=1}^K \tag{3}$$

Then, we utilize a prompt to instruct an LLM to transform the linearized data point to natural language, where the prompt consists of a prefix $p$ to describe the schema of the input features, the linearization, and a suffix $s$ (see the detailed prompts in Appendix A.1.1). Given the prompt, the LLM will generate a readable and concise natural description $z_{r,t}$, which can be formulated as:

$$z_{r,t} = \text{LLM}(p, \text{Linearize}(x_{r,t}), s) \tag{4}$$

It is worth noting that the missing variables will be replaced by a special token (e.g., [MASK]) for further imputation.

**Imputation of Incomplete Observations.** To handle the incomplete observation issue, we utilize a Sparse Mixture-of-Experts (SMoE) layer for imputing the incomplete observations. After transforming each data point into natural language description $z_{r,t}$, we feed it into a semantic time-series encoder $\mathcal{F}_l$ to get the encoded representations $\{m_{r,t}^h\}_{h=1}^H$ for each special token (i.e., incomplete feature), with $H$ denotes the number of missing variables. The encoding process can be formulated as follows:

$$\{m_{r,t}^h\}_{h=1}^H = \mathcal{F}_l(z_{r,t}) \tag{5}$$

Subsequently, to accommodate different characteristics of diverse physical variables, we apply a Sparse Mixture-of-Experts (SMoE) layer to impute the missing variables. Specifically, the encoded representation $\{m_{r,t}^h\}_{h=1}^H$ will be dynamically routed by an observation-independent noisy top-k gating network $\mathcal{G}$ to a subset of shared expert models $\{\mathcal{E}^e\}_{e=1}^E$ to facilitate the proper imputation of different observations. Following the original design of SMoE (Shazeer et al., 2017), the gating process can be formulated as:

$$\mathcal{G}(m_{r,t}^h) = \text{Softmax}(\text{TopK}(\mathcal{P}(m_{r,t}^h), k))$$
$$\mathcal{P}(m_{r,t}^h) = m_{r,t}^h \cdot W_g + \mu\text{Softplus}(m_{r,t}^h \cdot W_{noise}) \tag{6}$$
$$\text{TopK}(v, k)_j = \begin{cases} v_j, & \text{if } v_j \text{ is in the top k elements of } v \\ -\infty, & \text{otherwise} \end{cases}$$

where the $\mu$ is random noise sampled from a standard normal distribution, $W_g$ and $W_{noise}$ are learnable parameters. Then, the encoded representations $m_{r,t}^h$ will be routed only to the shared expert models $\{\mathcal{E}^e\}_{e=1}^E$ with top-k gating scores generated by the gating network $\mathcal{G}$. The predicted value $\widetilde{m}_{r,t}^h$ can then be calculated by combining the encoding results from the top-k expert models with their corresponding gating scores as:

$$\widetilde{m}_{r,t}^h = \mathcal{G}(m_{r,t}^h) \cdot \mathcal{E}(m_{r,t}^h) = \sum_{e=1}^E \mathcal{G}^e(m_{r,t}^h)\mathcal{E}^e(m_{r,t}^h) \tag{7}$$

Then, we can replace the incomplete observations in $z_{r,t}$ with the corresponding predicted values $\{\widetilde{m}_{r,t}^h\}_{h=1}^H$, obtaining the complete data $\widetilde{z}_{r,t}$.

**Multi-granularity information integration.** To address the distribution shift issue, we propose to incorporate multi-granularity information from previous observations. Specifically, for imputed time-series $\widetilde{z}_{r,t}$ in region $r$ at timestamp $t$, the incorporated multi-granularity information includes: (1) information from the past week in the same region, denoted as $\widetilde{z}_{r,t}^w = \{z_{r,t} \mid t \in \{t-1, \dots, t-7\}\}$; (2) information from the same day of each month of the year in the same region, denoted as $\widetilde{z}_{r,t}^y = \{z_{r,t} \mid t \in \{t-30, \dots, t-360\}\}$. Subsequently, we feed the current observation $\widetilde{z}_{r,t}$ into the semantic time-series encoder $\mathcal{F}_l$ to get the current embedding $U_{r,t}^c$. Likewise, weekly information $\widetilde{z}_{r,t}^w$ and yearly information $\widetilde{z}_{r,t}^y$ are encoded into weekly embedding $U_{r,t}^w$ and yearly embedding $U_{r,t}^y$, respectively. Finally, we concatenate all the multi-granularity information as the representation of semantic time-series to the LLM decoder, which can be formulated as:

$$U_{r,t} = U_{r,t}^c \oplus U_{r,t}^w \oplus U_{r,t}^y \tag{8}$$

where the $\oplus$ is the concatenation operation along the representation dimension.

## 3.2 Representation Learning from Temporal Trend Image

Aside from learning representations from semantic time-series, we also propose to capture the dynamics of physical variables in the temporal trend image. This image depicts the recent temporal dynamics of all variables simultaneously, making the capture of inter-variable relations more natural and easier. Specifically, we first convert the previous feature values into temporal trend images, which depict the temporal dynamics of all variables in the form of line graph images. Then, we utilize a vision encoder to encode the ST dynamics and correlations in the temporal trend image. These two steps are detailed below:

**Transforming environmental data into temporal trend image.** Given the current observation $x_{r,t}$, we utilize the line graph image to present each variables' (including the target variable's) observations in past $\beta$ days in the same region, i.e., $x_{r,t}^m = \{x_{r,t'} \mid t' \in \{t-1,\ldots,t-\beta\}\}$. For each variable in $x_{r,t}^m$, each point represents an observation identified by its time and value. As illustrated in Figure 2, for each variable (one subimage), the horizontal axis represents timestamps, and the vertical axis specifies values. Here, incomplete features are linearly interpolated, and the line graph images of all variables will be finally aggregated into one temporal trend image $i_{r,t}$. Following Li et al. (2024), we apply Z-normalization on all variables due to the different scales of environmental variables.

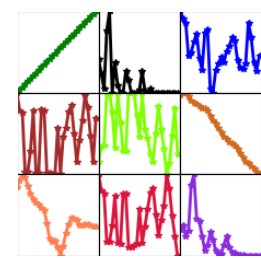

Figure 2: Illustration of temporal trend image.

**Image Recognition for temporal trend image.** After transforming the observations into the temporal trend image $i_{r,t}$, the temporal trend image $i_{r,t}$ is then embedded into a visual representation $O_{r,t}$ by the vision encoder $\mathcal{F}_i$, i.e., a pre-trained vision transformer (Liu et al., 2021b):

$$O_{r,t} = \mathcal{F}_i(i_{r,t}) \tag{9}$$

It is worth noting that the temporal trend image presents two types of information for vision transformer: (1) the temporal dynamics of a single variable in a line graph, and (2) the spatial relationships amongst the variables across different line graphs. The vision encoder can explicitly capture the inter-variable and intra-variable dependencies.

### 3.3 Inference with Large Language Models

After obtaining representations from semantic time-series and temporal trend image, we employ an LLM-based framework to fuse multimodal representations for prediction. In order to adapt the LLM to different applications, we propose to guide the frozen LLM with domain instructions. Concretely, the domain instruction consists of three parts: (1) dataset description, which provides essential background information of specific environmental application; (2) task description, which guides the LLM to perform multimodal fusion for the given task; (3) target statistics, which serves as crucial information for LLM to capture the recent dynamics of the target variable (see Appendix A.1.2 for detailed descriptions of domain instruction). For each data point $x_{r,t}$, we concatenate the learned representations from different modalities with domain instructions, and then forward them through a frozen LLM to obtain the output representation as:

$$Q_{r,t} = \text{LLM}(U_{r,t}, O_{r,t}, d_\alpha), \tag{10}$$

where $d_\alpha$ represents the domain instructions for a environment $\alpha$. Subsequently, we employ a linear layer on the top of output representation $Q_{r,t}$ to derive the final prediction $\widetilde{y}_{r,t}$. During training, we employ the prediction loss $\mathcal{L}_r$ and the imputation loss $\mathcal{L}_i$ of incomplete observations, which are defined as follows:

$$\mathcal{L}_r = \sqrt{\frac{1}{N} \sum_{r \in \mathcal{R}} \sum_{t \in \mathcal{T}} \|\widetilde{y}_{r,t} - y_{r,t}\|_2^2}, \quad \mathcal{L}_i = \sqrt{\frac{1}{M} \sum_{r \in \mathcal{R}} \sum_{t \in \mathcal{T}} \sum_{h \in \mathcal{H}_{r,t}} \left\|\widetilde{m}_{r,t}^h - m_{r,t}^h\right\|_2^2}, \tag{11}$$

where $N$ denotes the total number of observed target variables, $\mathcal{H}_{r,t}$ denotes the set of incomplete features in region $r$ at time $t$, and $M$ denotes the total number of incomplete features. The overall loss $\mathcal{L}_{all}$ is defined as follows:

$$\mathcal{L}_{all} = \eta_1 \mathcal{L}_r + \eta_2 \mathcal{L}_i \tag{12}$$

where $\eta_1$ and $\eta_2$ are coefficients to balance these loss terms. The whole algorithm for training is illustrated in Alg. 1 in Appendix A.2.

## 4 Related work

**Modeling of Environmental Ecosystems.** To address the environmental challenges, physics-based models have been commonly adopted in many areas such as hydrology and agriculture (Regan et al., 2018; Grant et al., 2010; Zhou et al., 2021), specifically like multi-purpose

terrestrial simulator  (Coon et al., 2019) and hydrological budget and crop yield predictor (Srinivasan et al., 2010). These models often rely on parameterizations and approximations due to incomplete knowledge or excessive complexity in modeling certain physical processes. As a result, they are often biased even after being calibrated using sufficient training data. As an alternative, advanced machine learning (ML) has been increasingly adopted to help understand the underlying complex physical processes and deal with large amounts of data. In particular, prior research has shown the promise of ML-based approaches in modeling agroecosystems (Jia et al., 2019; Licheng et al., 2021; Liu et al., 2022) and freshwater ecosystems (Read et al., 2019; Hanson et al.), and more recently, encouraging results by large language models in modeling environmental ecosystems (Luo et al., 2023). Both physics-based and ML-based models would benefit decision-making activities relevant to society, highlighting the intersection of environmental science, technology, and policy.

**Spatial-Temporal Prediction.** The ST prediction serves as a crucial topic in spatial-temporal data mining. Earliest models, e.g., ARIMA (Faruk, 2010), only focus on temporal dynamics. Later, multiple CNN-based and RNN-based models are proposed to simultaneously capture spatial-temporal correlations (Su et al., 2020; Sun et al., 2021; Yao et al., 2018; 2019b;a). Recently, with advanced deep learning methods such as graph neural networks and transformers, the performance on ST prediction has been significantly improved (Cini et al., 2023; Tang et al., 2023; Liu et al., 2023). However, ST prediction for environmental ecosystems still suffers from incomplete observations and distribution shifts.

**Imputation of Incomplete Observations.** One common issue in environmental data is the incomplete data, which is mainly caused by sensor failures and the high cost of measuring certain variables. To build a general framework for environmental ecosystems modeling, the first step is to properly handle the incomplete features. Previous research has proposed to address the incomplete observations by various data imputation methods (Berg et al., 2017; Gondara & Wang, 2018; Yoon et al., 2018; Spinelli et al., 2020) or by directly imputing the incomplete features when making decisions (Goodfellow et al., 2013; Śmieja et al., 2018; Ma et al., 2021; 2022; Tang et al., 2020). Nevertheless, these methods do not accommodate the heterogeneity amongst different variables in the incomplete observations, potentially leading to uninformed guessing when there are various missing values in one feature.

## 5   Experiments

In this section, we evaluate the performance of LITE, aiming to answer the following questions: **Q1:** Compared to previous methods, can LITE consistently perform well across different environmental science applications? **Q2:** Can the proposed modules (e.g., semantic time-series encoder) effectively improve performance? **Q3:** How does LITE perform when confronted with the challenge of incomplete features? **Q4:** Can LITE be robust to distribution shifts in environmental data?

### 5.1   Experimental Setup

**Datasets Descriptions.** In this subsection, we will briefly introduce three datasets from environmental ecosystems and Appendix A.3 provides further details.

- **CRW-Temp.** CRW-Temp is a stream water temperature prediction dataset, aiming to predict the daily average water temperature on a given day based on the observed physical variables on the same day. It is collected from the Christina River Watershed (CRW), which consists of 42 river segments. Water temperature observations are available for 22 segments on certain dates. The number of observations for each segment ranges from 1 to $3,572$ with a total of $12,506$ observations across all dates and segments. The temporal span of the dataset extends from October 31, 2006, to March 30, 2020, covering $4,900$ days. For the experiments, the first $2,450$ days are selected as the training split, while the subsequent $2,450$ days serve as the test split.

- **CRW-Flow.** The target of CRW-Flow is to predict the streamflow of river segments based on the observed physical variables. This dataset is also obtained from the Christina River

Table 1: Prediction RMSE and MAE for stream water temperature, stream flow, and agricultural nitrous oxide ($N_2O$) emission prediction. The best results are **bold**, while the second best results are underlined.

| Class | Method | CRW-Temp | | CRW-Flow | | AGR | |
|-------|--------|----------|--------|----------|--------|-----|-----|
| | | RMSE | MAE | RMSE | MAE | RMSE | MAE |
| Traditional | LSTM | $2.28 \pm 0.26$ | $1.82 \pm 0.20$ | $\underline{4.30 \pm 0.78}$ | $2.67 \pm 0.23$ | $0.51 \pm 0.12$ | $0.41 \pm 0.09$ |
| Graph-based | RGRN | $\underline{1.81 \pm 0.05}$ | $\underline{1.30 \pm 0.03}$ | $7.88 \pm 0.86$ | $2.80 \pm 0.10$ | $0.95 \pm 0.01$ | $0.76 \pm 0.03$ |
| | Gr-CNN | $1.86 \pm 0.14$ | $1.43 \pm 0.12$ | $8.77 \pm 0.10$ | $3.11 \pm 0.06$ | $0.34 \pm 0.03$ | $\underline{0.18 \pm 0.05}$ |
| | HydroNets | $1.87 \pm 0.31$ | $1.41 \pm 0.20$ | $7.21 \pm 0.35$ | $3.02 \pm 0.01$ | $\underline{0.18 \pm 0.01}$ | $0.21 \pm 0.01$ |
| | HRGN-DA | $1.87 \pm 0.01$ | $1.40 \pm 0.20$ | $6.49 \pm 0.56$ | $\underline{2.59 \pm 0.11}$ | $0.19 \pm 0.03$ | $0.22 \pm 0.01$ |
| Multimodal | LITE | $\mathbf{1.59 \pm 0.10}$ | $\mathbf{1.26 \pm 0.08}$ | $\mathbf{1.89 \pm 0.11}$ | $\mathbf{0.84 \pm 0.04}$ | $\mathbf{0.08 \pm 0.01}$ | $\mathbf{0.06 \pm 0.01}$ |

Watershed (CRW), while only 16 river segments in which have streamflow observations. The number of streamflow observations available for each segment ranges from 16 to $4,900$ with a total of $63,501$ observations across all dates and segments. The temporal span and dataset partition strategy of CRW-Flow are the same as CRW-Temp.

- **AGR.** AGR is an agricultural nitrous oxide ($N_2O$) emission prediction dataset acquired from a controlled-environment mesocosm facility in Minnesota, US. Six chambers were used to plant continuous corn during 2015–2018 and monitor the $N_2O$ emission hourly. The number of $N_2O$ emission observations available for each chamber ranges from $5,780$ to $6,168$ with a total of $35,711$ observations.

**Baselines.** For all datasets, we include five state-of-the-art baselines for environmental ecosystems modeling, including LSTM (Hochreiter & Schmidhuber, 1997), Graph-temporal convolutional network (Gr-CNN) (Sun et al., 2021), Physics-guided Recurrent Graph model (RGRN) (Jia et al., 2021), Hydronets (Moshe et al., 2020), and Heterogeneous Recurrent Graph Networks with data assimilation (HRGN-DA) (Chen et al., 2021).

**Implementation Details.** Following (Chen et al., 2021; Jia et al., 2021), we use RMSE and MAE to evaluate the performance of all methods. The default look-back window size in temporal trend image is set to 30 for all three datasets. Moreover, we utilize the pre-trained distilbert-base-uncased model (Sanh et al., 2019) as the semantic time-series encoder and the tiny Swin Transformer model (Liu et al., 2021a) as our vision encoder. The hidden dimension of the two pre-trained models is 768. Additionally, Llama-2-7b-hf (Touvron et al., 2023) is selected as the default backbone of our LLM decoder. For the sparsely-gated mixture-of-experts (SMoE) layer, each expert is a feed-forward network with hidden units of $\{768, 3072, 768\}$, the number of experts is 8 and top 2 experts will be selected by the router. We train our model via the AdamW optimizer with an initial learning rate of $0.00001$. All the experiments are implemented in PyTorch with one NVIDIA RTX A6000 GPU.

## 5.2 Main Results

Table 1 shows the performance of different approaches for predicting stream water temperature, streamflow, and agricultural nitrous oxide ($N_2O$) emission. Firstly, it can be observed that the graph neural networks-based methods, e.g., HRGN-DA(Chen et al., 2021), consistently outperforms the traditional approaches by a significant margin, demonstrating the importance of accurate modeling of non-stationary ST correlations. This observation is not surprising, since the distribution shifts in correlations of physical variables happen frequently, and will make learned models irrelevant. Second, LITE consistently outperforms the previous SOTA methods across all environmental domains, confirming its effectiveness in unifying environmental spatial-temporal prediction by representing spatial-temporal data from a multimodal perspective.

Table 2: Results of ablation study, which demonstrates the impact of different components on the overall performance of our model.

| | CRW-Temp | | CRW-Flow | | AGR | |
|---|---|---|---|---|---|---|
| | RMSE | MAE | RMSE | MAE | RMSE | MAE |
| LITE-text | $2.97 \pm 0.09$ | $2.53 \pm 0.09$ | $2.35 \pm 0.12$ | $1.20 \pm 0.04$ | $0.12 \pm 0.02$ | $0.10 \pm 0.01$ |
| LITE-image | $1.99 \pm 0.04$ | $1.66 \pm 0.04$ | $2.29 \pm 0.08$ | $0.96 \pm 0.03$ | $0.15 \pm 0.01$ | $0.13 \pm 0.01$ |
| LITE-LLM | $1.69 \pm 0.04$ | $1.44 \pm 0.04$ | $2.21 \pm 0.09$ | $0.97 \pm 0.03$ | $0.15 \pm 0.00$ | $0.13 \pm 0.00$ |
| LITE-imp | $1.65 \pm 0.09$ | $1.32 \pm 0.07$ | $2.04 \pm 0.14$ | $0.91 \pm 0.07$ | $0.14 \pm 0.02$ | $0.12 \pm 0.02$ |
| LITE-SMoE | $1.84 \pm 0.07$ | $1.54 \pm 0.06$ | $1.98 \pm 0.06$ | $0.88 \pm 0.04$ | $0.11 \pm 0.00$ | $0.14 \pm 0.00$ |
| LITE-mtg | $2.09 \pm 0.08$ | $1.76 \pm 0.07$ | $2.31 \pm 0.08$ | $0.93 \pm 0.06$ | $0.10 \pm 0.01$ | $0.09 \pm 0.00$ |
| **LITE** | $\mathbf{1.59 \pm 0.10}$ | $\mathbf{1.26 \pm 0.08}$ | $\mathbf{1.89 \pm 0.11}$ | $\mathbf{0.86 \pm 0.04}$ | $\mathbf{0.08 \pm 0.01}$ | $\mathbf{0.06 \pm 0.01}$ |

## 5.3 Ablation Studies

In this subsection, we conduct comprehensive ablation studies to demonstrate the effectiveness of our key modules, including the semantic time-series encoder, the temporal trend image, the SMoE for incomplete feature imputation, and the LLM decoder to perceive multimodal information. The ablation models include:

- **LITE-text**: In LITE-text, the semantic time-series encoder is removed and the representation from the vision encoder is directly fed into the LLM decoder.

- **LITE-image**: In LITE-image, the vision encoder is removed and the representation from the semantic time-series encoder is directly fed into the LLM decoder.

- **LITE-LLM**: In LITE-LLM, the LLM decoder is replaced with a linear layer.

- **LITE-imp**: In LITE-imp, the imputation module is removed. The special tokens (e.g., [MASK]) will not be imputed, leaving the input physical variables incomplete.

- **LITE-SMoE**: In LITE-SMoE, we replace the SMoE layer with a linear layer to impute incomplete features in semantic time-series encoder. Different characteristics of physical variable are not taken into account in the imputation step.

- **LITE-mtg**: In LITE-mtg, we do not incorporate multi-granular features from previous time in the language modality. Thus, data patterns of different granularities are not assimilated, which could hinder the ability to handle the common distribution shifts in environmental data.

The results are shown in Table 2, and the results of LITE are also reported for comparison. From this table, we observe that: (1) LITE outperforms all the variants without certain components, e.g., LITE-text, LITE-image, and LITE-LLM, which demonstrates the effectiveness and the complementary nature of time-series and temporal trend modalities. This also confirms the effectiveness of multiple components in LITE, including the imputation for incomplete features, the incorporation of multi-granularity information, and the domain-aware multimodal ST prediction with LLM. (2) LITE-SMoE performs worse than LITE, which implies the significance of using different experts for imputing different missing physical variables. Neglecting the heterogeneity among diverse variables will cast a negative impact on performance. LITE slightly outperforms LITE-imp, which shows the importance of imputing missing features. (3) LITE significantly outperforms LITE-mtg. This result is not surprising, as the incorporation of multi-granularity information indeed enhanced the model's ability to handle the temporal distribution shift.

## 5.4 Performance Under Leaving-Sensors-Out Settings

To evaluate our model in more challenging conditions, we conduct further experiments where a subset of features are missing during testing. This could commonly occur in real scenarios due to the difficulty in measuring certain physical variables or sensor malfunctions. Following the previous approach Zhang et al. (2022), we carried out assessments under two settings with the CRW-Temp dataset: (1) leave-fixed-sensors-out, which selects a fixed

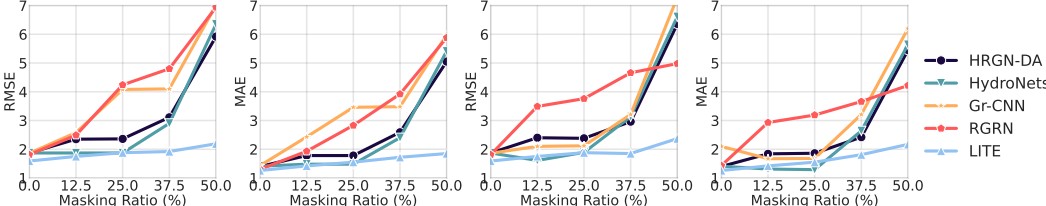

Figure 3: Experimental results under the leave-**fixed**-sensors-out (left two) and leave-**random**-sensors-out (right two) settings on CRW-Temp dataset.

set of sensors (i.e., features) and hides all their measured values in the test set, and (2) leave-random-sensors-out, which randomly selects sensors and hide all their measured values in the test set. See more detailed descriptions of both settings in Appendix A.4. In both settings, we set the missing ratio from 12.5% to 50.0%.

We present our results in Figure 3, which demonstrates our LITE model consistently outperforms all the baselines by a significant margin, especially under extreme conditions, e.g., when half of the physical variables are missing. We can observe that with the missing ratio increasing from 0% to 25%, most models' performance exhibits stability. However, under extreme conditions, i.e., 25%-50% of the features are missing, only our LITE model maintains strong performance, with the other methods showing significant performance drop (higher RMSE and MAE). Specifically, when increasing the missing ratio from 12.5% to 50%, the prediction RMSE of LITE only increases by 25%, while the average RMSE increase ratio of the other baselines is 185%. These results justifies the robustness of LITE to varying degrees of incomplete features, which are common in environmental ecosystem data.

### 5.5 Analysis on Domain Generalization Ability

In this subsection, we further test LITE to justify its robustness to distribution shifts in environmental data. Specifically, we often need to simulate environmental ecosystems across different regions, and these regions can exhibit diverse patterns due to their distinct system characteristics, e.g., soil and groundwater properties, weather conditions, etc. In this test, we aim to generalize the model to test domains (regions) $\mathcal{D}^{test}$ that are disjoint from the training domains (regions) $\mathcal{D}^{train}$, i.e., $\mathcal{D}^{train} \cap \mathcal{D}^{test} = \varnothing$. For example, on the CRW-Temp dataset, we train each

| Method | RMSE | MAE |
|---|---|---|
| LSTM | $7.10 \pm 0.06$ | $5.90 \pm 0.06$ |
| RGRN | $2.71 \pm 0.03$ | $2.08 \pm 0.04$ |
| Gr-CNN | $2.41 \pm 0.20$ | $2.30 \pm 0.16$ |
| HydroNets | $1.85 \pm 0.12$ | $1.52 \pm 0.03$ |
| HRGN-DA | $1.73 \pm 0.16$ | $1.33 \pm 0.10$ |
| LITE | $\mathbf{1.41 \pm 0.23}$ | $\mathbf{1.31 \pm 0.18}$ |

Table 3: Prediction RMSE and MAE under out-of-distribution regions.

model on three river segments that have the most observations and test on the remaining river segments. We present the experimental results in Table 3. It can be seen that our LITE shows superiority under out-of-distribution (OOD) testing. This result verifies LITE's robustness to distribution shift that frequently happens in environmental ecosystems, demonstrating its potential as a general framework for environmental ST prediction.

## 6 Conclusion

This paper presents LITE, a multimodal Large Language Model for environmental Ecosystem Modeling. By transforming the spatial-temporal data into semantic time-series and temporal trend images, we could effectively learn multimodal representations of environmental data from pre-trained foundation models. We employ a Sparse Mixture-of-Experts (SMoE) layer to impute the incomplete observations and incorporate multi-granularity information to handle the distribution shifts in environmental data. Additionally, we utilize domain instructions to guide a frozen LLM to fuse multimodal representation. Our extensive experiments demonstrate the effectiveness and robustness of our LITE method, particularly with varying degrees of missing features and distribution shifts.

## Acknowledgement

We thank Google Cloud Research Credits program for supporting our computing needs.

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

# A    Appendix

## A.1    Prompt

We provide detailed descriptions and examples of prompts used in this paper.

### A.1.1    Prompt to Guide Semantic Time-series Generation

In Sec 3.1, we utilize a prompt to instruct an LLM to transform the linearized data point to natural language, where the prompt consists of: a prefix p to describe the schema of the input features, the linearization and a suffix s. We detail the prompt below.

Table 4: Prompt for generating semantic time-series.

| |
|---|
| **System Message** |
| You are a helpful assistant. |
| **Prompting** |
| Here is the schema definition of the table: $schema_definition |
| This is a sample from the table: $linearization |
| Please briefly summarize the sample with its value in one sentence. |
| You should describe the important values, like rainfall and average cloud cover fraction, instead of just the names of columns in the table. |
| A brief summarization of another sample may look like: "On November 14, 2009, the rainfall in the Delaware River Basin was 0.00152 inches. The average air temperature was [MASK] degrees Celsius. The solar radiation recorded was 125.31325 units. The average cloud cover fraction was 0.4352." |
| Note that the example is not the summarization of the sample you have to summarize. |
| **Response** |
| $ semantic_time-series_of_the_given_sample |

### A.1.2    Domain Instruction

Here, we provide two examples for the domain instruction we used to guide the frozen LLM, which consists of three parts: dataset description, task description, and target statistics.

- $<|start\_prompt|>$ **Dataset description:** The Christina River Watershed Flow (CRW-Flow) is a dataset containing streamflow observations from 16 river segments. It is worth noting that the streamflow becomes hundreds of times higher than usual when it rains. **Task description:** predict the stremflow given the observed meteorological features represented in the image and text spaces; **Target statistics**: min value {min_values_str}, max value {max_values_str}, median value {median_values_str}, the trend of input is {trend} $<|<end\_prompt>|>$

- $<|start\_prompt|>$ **Dataset description:** The Agriculture nitrous oxide (AGR) is a dataset containing agricultural nitrous oxide emission observations from 6 chambers. **Task description:** predict the nitrous oxide emission given the observed meteorological features represented in the image and text spaces; **Target statistics**: min value {min_values_str}, max value {max_values_str}, median value {median_values_str}, the trend of input is {trend} $<|<end\_prompt>|>$

### A.2 Pseduocodes of training algorithm

The pseuduocodes of training algorithm is shown in Alg. 1.

---
**Algorithm 1** Training algorithm of LITE

1: **Input:** Training dataset $\mathcal{D} = \{x_{r,t}, y_{r,t}\}_{t=1}^{T}$
2: *step 1. Transforming the data into semantic time series and temporal trend image;*
3: **for** $t \leftarrow 1$ **to** $T$ **do**
4:     $z_{r,t} \leftarrow LLM(p, Lineariza(x_{r,t}), s)$
5:     $i_{r,t} \leftarrow x_{r,t}^{m}$
6: **end for**
7: *step 2. Train LITE;*
8: **for** *each minibatch* $\mathcal{B}$ in dataset $\mathcal{D}$ **do**
9:     **for** $t \leftarrow 1$ **to** batch size $b$ **do**
10:       impute the missing variables with SMoE and obtain a complete semantic time-series $\tilde{z}_{r,t}$ as Equation (4), (6), (7)
11:       $U_{r,t} \leftarrow \mathcal{F}_{l}(\tilde{z}_{r,t}) \oplus \mathcal{F}_{l}(\tilde{z}_{r,t}^{w}) \oplus \mathcal{F}_{l}(\tilde{z}_{r,t}^{y})$
12:       $O_{r,t} \leftarrow \mathcal{F}_{i}(i_{r,t})$
13:       Fuse multimodal representations as Equation (10)
14:     **end for**
15: **end for**
16: Compute the losses $\mathcal{L}_{r}, \mathcal{L}_{i}$ and $\mathcal{L}_{all}$ following Equation (11), (12)
17: Optimize the parameters $\theta$ of LITE by minimizing $\mathcal{L}_{all}$

---

### A.3 Dataset Description

In this subsection, we provide detailed descriptions for the three datasets we used in experiments, i.e., CRW-Temp, CRW-Flow, and AGR.

- **CRW-Temp.** This is a stream water temperature dataset originating from the Delaware River Basin, an area recognized for its ecological diversity and as a significant watershed on the United States' east coast. It originates from the U.S. Geological Survey's National Water Information System (Survey, 2016) and the Water Quality Portal (Read et al., 2017), which is the most extensive standardized dataset for water quality in both inland and coastal waters. The data focuses on specific geographic coordinates, aligning observations with river segments that range in length from 48 to 23,120 meters. These segments are delineated according to the national geospatial framework utilized by the National Hydrologic Model (Regan et al., 2018). To ensure observations are accurately associated with the corresponding river segments, we align observations to the nearest stream segment within a 250-meter tolerance. It is worth noting that: 1) For segments with more than one observation site, data was consolidated into a singular daily average water temperature; 2) Observations located more than 5,000 meters along the river channel

from the end of a segment were excluded from the dataset. Our study focuses on specific portions of the Delaware River Basin, particularly those that merge into the main stem of the Delaware River in Wilmington, DE. We refer to this large dataset as the Christina River Watershed (**CRW**), which includes 42 river segments. The temporal span of the dataset extends from October 31, 2006, to March 30, 2020, covering $4,900$ days. For a given region $r$ and date $t$, the daily meteorological features are: *the day of the year, rainfall, daily average air temperature, solar radiation, average cloud cover fraction, groundwater temperature, subsurface temperature, and potential evapotranspiration.* Considering that all physical variables (both features and targets) are sourced from sensors, they display various levels of sparsity. Overall, the dataset contains $12,506$ valid temperature observations.

- **CRW-Flow.** CRW-Flow is a streamflow prediction dataset that is also collected on the Christina River Watershed (**CRW**). The temporal span, meteorological features, and all other default settings of CRW-Flow are the same with CRW-Temp. The only difference is that the target of CRW-Temp is to predict streamflow instead of stream water temperature. There are in total $63,501$ valid observations of streamflow in CRW-Flow.

- **AGR.** AGR is an agricultural nitrous oxide ($N_2O$) emission prediction dataset acquired from a controlled-environment mesocosm facility in Minnesota, US. Six chambers with a soil surface area of 2 $m^2$ and column depth of 1.1 m were used to plant continuous corn during 2015–2018 and monitor the $N_2O$ emission hourly ($mgNm^{-1}h^{-1}$) with a $N_2O$ analyzer (Teledyne M320EU, Teledyne Technologies International Corp, Thousand Oaks, CA). The number of $N_2O$ emission observations available for each chamber ranges from $5,780$ to $6,168$ with a total of $35,711$ observations. We apply Z-normalization to the data from AGR before feeding it into our model.

### A.4   Leave-Sensors-Out Settings

Here, we provide detailed description of two leave-sensors-out settings we adopt on CRW-Temp dataset, i.e, leave-**fixed**-sensors-out and leave-**random**-sensors-out.

**Leave-fixed-sensors-out.** For a fair comparison, we hided the same set of sensors across all methods. Specifically, we selected 4 features from CRW-Temp, which are rainfall, average cloud cover fraction, groundwater temperature, and subsurface temperature. The selection process was incremental, starting with the exclusion of one feature and gradually expanding to encompass all four.

**Leave-random-sensors-out.** Under this setting, features in the test set are randomly dropped to introduce variability. Same with the leave-fixed-sensors-out setting, we will start with dropping one random feature, and finally drop four random features. This stochastic feature omission simulates real-world scenarios where data may be missing unpredictably, thus providing further insights into our model's adaptability under less controlled conditions.

