# OpenReview forum: "LITE: Modeling Environmental Ecosystems with Multimodal Large Language Models"
_colmweb.org/COLM/2024/Conference — COLM_

### Official Review · Reviewer_CCyC · 2024-04-30

**Rating:** 7
**Confidence:** 3
**Ethics Flag:** 1

**Summary:**

This work proposed a multi-model LLM framework for modeling time series data.
Two different representations 1) text description 2) trend image and a domain description are combined to represent each instance for prediction. In order to deal with various challenges in the environmental ecosystem data 1)  incomplete features 2) distribution shifts, mixture of experts and multi-granularity features are used.

Experiment on three datasets (water temp, stream flow, agricultural N2O) shows significant improvement over various SOTA approaches. Ablation study shows the importance of each component and especially the text description, the trend image and multi-granularity features.

**Reasons To Accept:**

This work shows the importance of combining image and text representation for modeling time series data. It can seem as an extension to the ViTST approach which only leverage the trend image representation.

**Reasons To Reject:**

This is a fairly complex system. Although an ablation study is provided, I still feel that the paper can be stronger if alternative solution for each challenge can be discussed and compared to.

---

> ### Author Rebuttal · Authors · 2024-05-30
>
> Thank you for your valuable and insightful reviews.
>
> In our ablation studies, we have explored some simplified versions of LITE, e.g., replacing the frozen LLM with a linear layer (LITE-LLM), and removing the multi-granularity information (LITE-mtg). However, such changes will compromise some performance degradation. From the experimental results, we observe the most necessary parts aside from language including vision (temporal trend image), multi-granularity information to handle distribution shifts, and imputation to handle incomplete observations. The other designs, e.g., the SMoE layer for imputation, can be removed or replaced under some circumstances. Our results in ablation studies can provide some useful guide for future research or applications.

---

> > ### Comment · Reviewer_CCyC · 2024-06-05
> > **Thanks for adding more details**
> >
> > Thanks for adding more details

---

### Official Review · Reviewer_DnLc · 2024-05-11

**Rating:** 6
**Confidence:** 3
**Ethics Flag:** 1

**Summary:**

For handling incomplete features and distribution shifts, this paper proposes Multimodal Large Language Model for EnvIronmenTal
Ecosystem Modeling (LITE) to better capture and process ST dynamics and correlations from a multimodal perspective. The experiments
demonstrate that LITE significantly enhances performance in environmental spatial-temporal prediction.

**Questions To Authors:**

1. Why dose transform the linearized data point to natural language?
2. How to find missing variables after transforming the linearized data point to natural language?
3. What is the role of imputation loss？What if delete it？

**Reasons To Accept:**

1. The proposed method is effective;

**Reasons To Reject:**

1. The  latest baselin is in 2021, which is relatively old. Why are there no updated baselines?
2. In ablation Studies, because the difference between the parameters of a linear layer and Llama-2-7b is too great, the effect of LLM, the role of LLM is not prominent enough. If the LLM decoder is replaced with small LM, it can better reflect the role of LLM.
3. In Multi-granularity information, The contribution of the two types of data, week data and year data, is unclear.

---

> ### Author Rebuttal · Authors · 2024-05-30
>
> Thank you for your valuable reviews and we have conducted experiments according to your suggetions.
>
> **Latest baseline.** While the modeling of environmental ecosystems is crucial to people’s livelihood, this area is under-explored. We summarize the past efforts at the accurate modeling of environmental ST data and propose a novel multimodal LLM-based method that can serve as a new strong baseline and provide useful insight for future research.
>
> **Replace LLM decoder with LM.** We have conducted experiments using a smaller LM - distilbert, to replace the frozen LLM (LITE-LM).
>
> |  | CRW-Temp |  | CRW-Flow |  | AGR |  |
> | --- | --- | --- | --- | --- | --- | --- |
> |  | RMSE | MAE | RMSE | MAE | RMSE | MAE |
> | LITE-LM | 1.98 | 1.57 | 2.96 | 1.11 | 0.15 | 0.14 |
> | LITE | 1.59 | 1.12 | 1.89 | 0.86 | 0.08 | 0.06 |
>
> Although we tune the LM, it performs worse than the LLM version. This is due to its limited ability to fuse the multimodal representations under domain instructions. In contrast, the LLM in LITE is frozen and performs much better.
>
> **Week and year data.** In LITE-year, we keep only the week data. In LITE-week, we keep only the year data. We also include results of LITE-mtg (remove week and year data) and LITE for comparison.
>
> |  | CRW-Temp |  | CRW-Flow |  | AGR |  |
> | --- | --- | --- | --- | --- | --- | --- |
> |  | RMSE | MAE | RMSE | MAE | RMSE | MAE |
> | LITE-year | 1.65 | 1.33 | 2.02 | 0.88 | 0.09 | 0.08 |
> | LITE-week | 1.87 | 1.53 | 2.29 | 0.92 | 0.13 | 0.11 |
> | LITE-mtg | 2.09 | 1.76 | 2.31 | 0.93 | 0.10 | 0.09 |
> | LITE | 1.59 | 1.12 | 1.89 | 0.86 | 0.08 | 0.06 |
>
> Our observations are: (1) both week and year data contribute to the superior performance of LITE; (2) week data is of greater significance than year data in terms of the above benchmarks. (3) combing week and year data, which is our multi-granularity information, yields best performance.
>
> **Linearized data to natural language?** Linearized data lacks the smoothness of natural language, which is important for modeling the inter-variable correlations in the semantic space.
>
> **How to find missing variables?** For any missing variable, like air temperature, we represent it as “air temperature: None” in the linearized data. After transforming it into natural language, we replace “None” with a special token, [MASK], for further imputation.
>
> **The role of imputation loss** is to guide the imputation of missing variables. When we remove it in LITE-imp, this lead to a performance drop.

---

### Official Review · Reviewer_cTdF · 2024-05-14

**Rating:** 7
**Confidence:** 3
**Ethics Flag:** 1

**Summary:**

This paper proposes a new hybrid arch to model environmental ecosystems, predicting physical variables such as water temperatures in a given region at a specific time step. The authors first prompt LMs to transform input features into natural language descriptions, then employ a semantic time-series encoder to encode these descriptions. These text features are fused with image features as well. The arch allows for imputation of missing features and the authors further integrate multi-granularity information to mitigate distribution shift issues. Finally, all these features are fed into a frozen llama LLM to make the final prediction. Comprehensive experiments and ablation analysis demonstrate the effectiveness of the approach.

**Reasons To Accept:**

1. Applying LLMs to spatial-time series data prediction in the environmental context is very interesting and potentially impactful for society. The authors fuse both text and image features that are novel.
2. The main results are quite good, greatly improving previous baselines that are kind of traditional.
3. While many designs zoom into specific details, the authors try to conduct comprehensive ablation analysis to justify the effectiveness of each component. For example, I appreciate the ablation experiments that focus on the settings with missing features and distribution shifts.

**Reasons To Reject:**

I don’t have any significant concerns for this paper.

---

> ### Author Rebuttal · Authors · 2024-05-30
>
> Thank you for your valuable reviews and for acknowledging the novelty and effectiveness of our approach.

---

### Official Review · Reviewer_hAX4 · 2024-05-14

**Rating:** 7
**Confidence:** 3
**Ethics Flag:** 1

**Summary:**

The authors present LITE, a multimodal Large Language Model for environmental Ecosystem
Modeling. LITE uses spatial-temporal data and transforms these into two data modalities: semantic time-series and
temporal trend images. These transformed data are then encoded to learn multimodal representations of environmental data from pre-trained foundation models. The work addresses gaps frequently arising in real-world data: by imputing the incomplete observations; and incorporating multi-granularity
information to handle data distribution shifts in environmental data.
A decoder, in the form of a frozen LLM, is used to fuse these multimodal representations. The authors demonstrate the feasibility and generalizability of LITE on real-world environmental data.

The motivation, ideas and previous literature are well discussed, but there are points that need further clarification.

**Questions To Authors:**

The motivation, ideas and previous literature are well discussed, but there are points that need further clarification. In general; a) the functional forms of Equations are missing; and b) a table with variables and their usage/function in the Appendix will be a good reference since there are a handful of variables that are not defined.

Please note these points below.

1.	Page 2: 2nd paragraph. Please provide examples of ‘variables’, and therefore ‘inter-variable correlations’. One could use the same environmental application to provide this.
2.	Eqns 3, 5, 9; what are these functions? Provide a functional form or a reference.
3.	Comparing Eqns 3 and 2: Is \x_{r,t} the same as a data point or feature value or physical driver or an observation? Can these terms be used interchangeably? Provide additional clarification on these definitions.
4.	Another term that caused confusion is ‘missing variables’. On page 4, the first line mentions that missing variables are replaced. After Eqn 5, there is mention that missing variables are imputed using SMoE, as well as in the Algorithm 1. If both procedures are done, then update the text and Algorithm accordingly. Please clarify.
5.	Eqn 6. What is v (parameter to TopK)?
6.	Eqn 7. What is E?
7.	Figure 2: needs additional details. Is this a single i_{r,t} or a group of them shown?
* What variable is being plotted
* Label what each subplot is
* what the x-axis and y-axis are
* are the axes the same for each subplot
* is {\beta} = 9 which is why there are 9 subplots
* are these z-normalized values?
8.	What is the difference between the LLMs in Eqn 4 and 10
9.	The algorithm is still incomplete, for example, it does not mention Eqn 10 or steps needing Eqn 10.
10.	Line immediately after Eqn 10: ‘we employ a linear layer on the top of output representation….”. What is the functional form of this layer, and why was it needed?
11.	What is the role of \eta_1 and \eta_2 in the Loss function?
* What does it mean to ‘balance loss terms’?
* Do these have constraints?
* where there any bounds to the loss function?
12.	Legend in Fig 3 is not clear. Which curves relate to the fixed versus random sensors?
13.	Is the average RMSE increase in ratio ‘223%’?

**Reasons To Accept:**

The paper is well written, but does need tidying up. Please see comments.

**Reasons To Reject:**

None

---

> ### Author Rebuttal · Authors · 2024-05-30
>
> Thank you for your valuable and detailed reviews.
>
> **Variables** in the CRW-Temp dataset includes rainfall, air temperature, etc. For instance, the rainfall on a given day can influence the air temperature.
>
> **Eqn 3, 5, 9.** Example of Eqn 3: time: 2017-01-07, radiation: 0.5, air temperature: 25, precipitation: 1.2, spRH: 0. For Eqn 5, please refer to the fourth to second lines before it. In Eqn 9, we encode the temporal trend image with a swin transformer.
>
> $\mathbf{x_{r,t}}$ is a data point, which has K features.
>
> **Notations about ‘missing variables’**. Please refer to the first sentence after Eqn 7, and lines 4 and 10 of Algorithm 1.
>
> **V in Eqn 6** is the input to the TopK function, i.e., the representation of each missing variable.
>
> **E in Eqn 7** is the shared expert models which consists of E expert models.
>
> **Figure 2** is an example of a single $i_{r,t}$ in the CRW-Temp dataset, which contains the observations of 9 variables over the past 30 days ($\beta=30$). From left to right and top to bottom, the variables follow the same order as listed in lines 14 to 16 on page 14. The x-axis represents timestamps for all subplots, while the y axis specifies the values of each variable in a subplot. The units of the y-axis in different subplots are different, so we apply z-normalizaition on all variables before plotting the temporal trend image.
>
> **LLMs in Eqn 4 and 10** are GPT-3.5-turbo and LLaMA2-7b, respectively. We will modify the notation of LLM in Eqn 4 in the future version.
>
> **Eqn 10 in the algorithm** is in line 13.
>
> **Linear layer after Eqn 10** is: $(Dropout(Q_{r, t}W_{1}+b_{1}))W_{2}+b_{2}$. It is used to transform the output representation from the LLM ($d=d_{LLM}$) to the final prediction ($d=1$).
>
> $\eta_{1}$  **and** $\eta_{2}$ **in the loss function** are used to balance the regression loss and the imputation loss, which is much higher than the former one. The hyperparameters of $\eta_{1}$ and $\eta_{2}$ are decided through cross validation. This is a very common practice in literature (e.g., [1, 2]).
>
> [1] Sener et al. “Multi-task learning as multi-objective optimization.” NeurIPS’18.
>
> [2] Bhattacharjee et al. “Mult: An end-to-end multitask learning transformer.” CVPR’22.
>
> **Figure 3.** From left to right, the first two are fixed settings while the second two are random settings.
>
> **Average RMSE increase ratio** of the baselines is: $[(5.92-2.35)/2.35+(6.35-1.87)/1.87+(7.01-2.58)/2.58+(6.94-2.49)/2.49]/4=185$%. We will fix this typo.

---

### Decision · Program_Chairs · 2024-07-10

**Decision:**

Accept

**Comment:**

This paper introduces a multimodal LLM-based model for modeling environmental ecosystems, characterized by sequential data (e.g. series of observations and predictions over time), missing data, and multiple modalities (here discrete numerical observations that are transformed into images depicting trends and into natural language).  Reviewers generally found this work be well written and experimentally robust, with good ablation studies disentangling the important modeling contributions.  Overall this is solid work that illustrates how numerical data can be transformed into text and images to improve predictive accuracy.